# Unusual Masquerading Infraorbital Mass—A Case Report of Human Ocular Dirofilariosis Found in A Ukrainian Patient in Switzerland

**DOI:** 10.3390/pathogens12080982

**Published:** 2023-07-27

**Authors:** Feliciana Menna, Filippo Billia, Anthia Papazoglou, Markus Tschopp, Rainer Grobholz

**Affiliations:** 1Department of Ophthalmology, Cantonal Hospital Aarau, CH-5001 Aarau, Switzerland; filippo.billia@gmail.com (F.B.); anthia.papazoglou@ksa.ch (A.P.); markus.tschopp@ksa.ch (M.T.); 2Department of Ophthalmology, Inselspital, Bern University Hospital, University of Bern, 3010 Bern, Switzerland; 3Medical Faculty, Institute of Pathology, Cantonal Hospital Aarau, University of Zurich, CH-5001 Aarau, Switzerland; rainer.grobholz@ksa.ch; 4Institute of Pathology, Cantonal Hospital Aarau, CH-5001 Aarau, Switzerland

**Keywords:** infraorbital cyst, subcutaneous dirofilariosis, *Dirofilaria repens*, orbital tumor

## Abstract

A 78-year-old Ukrainian woman who had immigrated to Switzerland presented with a rapid growing subcutaneous infraorbital mass. Surgical excision of the mass revealed a well-circumscribed, encapsulated tumor, adherent to the skin. The excision showed a soft tissue inflammation with parts of *Dirofilaria* spp. The number of cases of human dirofilariosis reported in the last 50 years has gradually increased. *Dirofilaria repens* is now endemic in many countries and is currently considered to be one of the fast spreading zoonoses in Central, Eastern and Northern Europe. The first empirical evidence of Swiss spreading of *D. repens* infections was in a dog from southern Switzerland in 1998. Ours is the first case of human orbital dirofilariosis found in a Ukranian patient reported in Switzerland. Our purpose is to inform the ophthalmologist to consider orbital dirofilariosis in the differential diagnosis of inflammatory masses of the orbit and to warn about the spread of this infection in Switzerland.

## 1. Introduction

A wide variety of processes can produce space-occupying lesions in and around the orbit. These include benign neoplasms, malignant neoplasms, vascular lesions, inflammatory disease, congenital lesions, and infections, among other causes. Imaging can be used to precisely localize a lesion, to help establish a diagnosis or generate a differential diagnosis that guides management, to follow a known lesion for progression, or some combination of these [1].

In this report we describe a case of a rapidly growing subcutaneous infraorbital mass—masquerading as orbital tumor—secondary to a dirofilariosis.

This case report was prepared in accordance with the principles of the Declaration of Helsinki. As this article pertains only to a case report, specific ethics approval is not mandated. Written informed consent was taken from the patient to publish the photographs before inclusion in the article.

## 2. Case Report

A 78-year-old Ukrainian woman, who had moved to Switzerland about 6 months ago complained of a three-week history of sudden, relatively painless swelling in the lower right orbit (Figure 1). At the time of presentation in our clinic in November she had already been treated with corticosteroids and antibiotics, with no improvement. Since the patient had been recently diagnosed with breast cancer, she was referred to us with the suspect of an orbital tumor or metastasis. She had no dogs of her own but lived in a family with dogs in Switzerland. She reported no recent foreign travel and could not recall any mosquito bites. Vision was 20/20 in both the eyes and intraocular pressure and fundus were normal. There was no evidence of conjunctival infection and hemorrhage. A round palpable hard and not very mobile subcutaneous mass was in the area of the lower lid/ anterior orbit. CT scan showed a round well-defined mass in the lower left orbit (Figure 2). An excision was performed under local anesthesia. The histological evaluation section showed a florid nodular soft tissue inflammation measuring 1.4 × 1 × 1 cm with abundant neutrophils and eosinophils. Centrally, sections of a nematode parasite could be observed. The nematode had an average cross-section of 450 μm with an approximately 7 μm thick cuticle with typical longitudinal grooves (Figure 3A,B). Within the worm, parts of the intestine and a paired uterus were visible. Based on this morphology, the parasite was a female *Dirofilaria* spp., probably *Dirofilaria repens.* The postoperative course went well and the patient had no complaints after the surgical excision of the mass.

## 3. Discussion

*Dirofilaria repens* is a nematode affecting domestic and wild canids, transmitted by several species of mosquitoes. It usually causes a non-pathogenic subcutaneous infection in dogs and is the principal agent of human dirofilariosis in Europe [2]. In recent decades, *D. repens* has increased in prevalence in areas where it has already been reported and its distribution range has expanded into new areas of Europe, representing a paradigmatic example of an emergent pathogen [3].

In the ophthalmology literature, the number of cases of human dirofilariosis reported in the last 50 years has gradually increased. Global warming in particular, has created conditions favoring the development of infective larvae in mosquitoes, and facilitated the recent spread of *Dirofilaria* spp. to Central Europe. *Dirofilaria repens* is now endemic in many countries in the region (Poland, Ukraine, Germany, Austria, Hungary, The Netherlands), and is currently considered to be one of the fastest spreading zoonoses in Central, Eastern, and Northern Europe [4,5]. In Ukraine, reporting cases of dirofilariasis has been mandatory since 1975, and the disease was included in the national surveillance system for notifiable diseases [6].

The first empirical evidence of Swiss spreading of *D. repens* infections was found in a dog in southern Switzerland in 1998. A few years later, another two positive dogs were found in Canton Ticino in 2001 [7,8].

*Dirofilaria repens* is a common parasite in dogs, which constitute the main source of infection. Humans are accidental hosts and many infected subjects are asymptomatic [5]. Transmission occurs through the bite of zooanthropophilic species of the mosquito genera *Aedes*, *Culex*, or *Anopheles* mosquitoes carrying infective larvae acquired from the microfilariae-rich blood of animal hosts parasitized with either deep-seated or subcutaneous worms of the *Dirofilaria* spp. In humans, the nematode causes a subcutaneous or superficially located inflammatory reaction that traps it within a nodule, where it may survive for many years [9]. Such lesions are always associated with moderate to severe inflammation. However, it may also present as a noninflammatory lid tumor.

*Dirofilaria repens* is well known to affect the eye and the ocular adnexa [10]. The infection may be periocular, subconjunctival, or intraocular. The first case of ocular dirofilariosis was reported in 1885 by Addario [11], an Italian ophthalmologist. He found a *Dirofilaria* worm in the conjunctiva of a Sicilian woman and called it *Dirofilaria conjunctivae*. The largest series of cases–six with periocular involvement of *D. repens*—was described by Font in 1980 [12]. More recently reports have included one involving the lateral rectus muscle of a 20-year-old man [13], conjunctival tissue of a 27-year-old female [14] and a 35-year-old male [15] and a superficial orbital dirofilariosis in a 24-year-old female [16]. A case was recently reported of ocular dirofilariosis in a 76-year-old patient in the course of cataract surgery in Bulgaria caused by a gravid female nematode [17] and a case of a 12-year-old patient with a rapidly growing deep orbital mass and imaging findings suggestive of rhabdomyosarcoma that was found to be dirofilariosis after mass resection [18]. Our case showed the typical morphologic picture of a female *D. repens*; however, based on histologic criteria alone, in the differential diagnosis an onchoceriasis has to be considered. *Onchocerca lupi* (zoonotic) is also present in Europe. In the present case, the morphology of the parasite makes us think with high probability of an infection with *D. repens*; the morphological distinctive characters of female nematodes allow a straightforward identification. *Onchocerca lupi* presents a thick cuticle composed of an external layer bearing prominent, undulated annular ridges and an internal layer with transverse striae [19]. These morphological features are not present in *D. repens*, which has longitudinal crests. Since *O. volvulus* occurs mainly in Africa, with additional foci in Latin America and the Middle East, the present case represents, with a very high probability, *D. repens* infection rather than an onchocerciasis.

In the literature there are 38 up-to-date reported cases of ocular/orbital/eyelid dirofilariosis. The majority of articles were case reports of up to three patients, with the exception of Dzamic et al. [20] who presented 19 cases of human dirofilariosis in Serbia, both ocular and subcutaneous, and Kalogeropoulos et al. [21], who presented eight cases of ocular dirofilariosis. In published reports of ocular dirofilariosis, most of the cases were located under the conjunctiva (>60% of all cases), followed by orbital/eyelid dirofilariosis (approximately 25%).

In general, the diagnosis of human dirofilariosis is based on histologic examination. Useful characteristics for differentiating between the different *Dirofilaria* spp. are the size and the features of the body wall, i.e., thickness of the cuticle and its structure, ridges, lateral chords, and number and type of muscle cells [22].

Usually, the clinical symptoms disappear after the parasite is removed and no adjunct therapy is necessary [23].

## 4. Conclusions

This is the first report of human infraorbital dirofilariosis masquerading as an orbital tumor found in a Ukrainian patient reported in Switzerland. 

Climate change affecting mosquito vectors and the facilitation of pet travel seem to have contributed to this expansion; however, the major factor is likely the rate of undiagnosed dogs perpetuating the life-cycle of *D. repens* [2]. Many infected dogs remain undetected due to the subclinical nature of the disease, the lack of rapid and reliable diagnostic tools and the poor knowledge and still low awareness of *D. repens* in non-endemic areas.

Because the clinical onset of orbital dirofilariosis is usually marked by common symptoms of orbital inflammation such as redness and swelling, ophthalmologists are advised to consider this parasite in the differential diagnosis of inflammation of the orbital tissues. The differential diagnosis in these cases incudes dermoid cysts, sarcoidosis, idiopathic pseudotumor, benign and malignant tumors including metastatic disease.

The purpose of this report is to inform ophthalmologists to consider orbital dirofilariosis in the differential diagnosis of inflammatory masses of the orbit and to warn about the spread of this infection in Switzerland.

In our view, it is critical to register all new cases in order to assess the epidemiological status and raise awareness of this rare infection.

## Figures and Tables

**Figure 1 pathogens-12-00982-f001:**
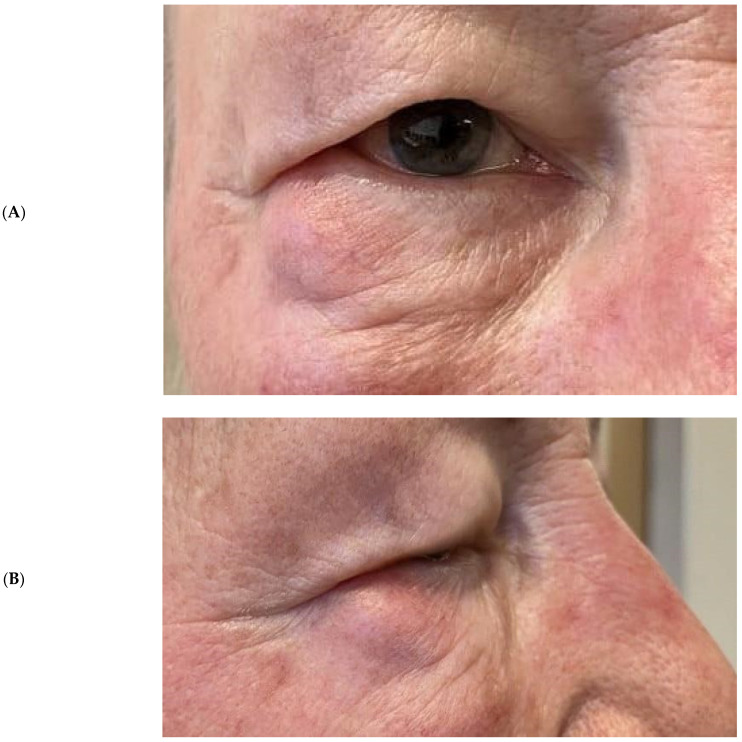
(**A**,**B**) This 78-year-old woman had three weeks history of sudden and non-specific swelling in the lower right orbit.

**Figure 2 pathogens-12-00982-f002:**
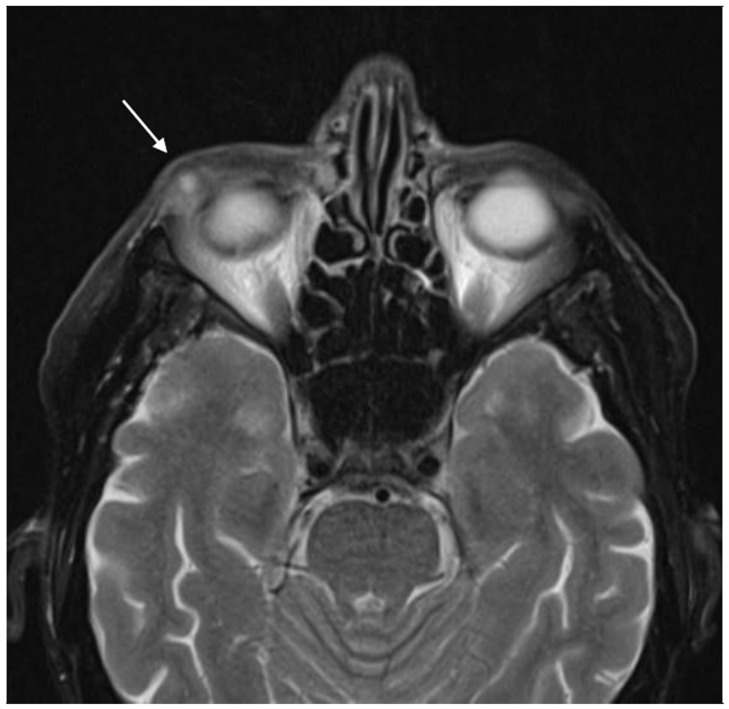
A CT scan showed a round, well-defined mass in the lower right orbit (arrow).

**Figure 3 pathogens-12-00982-f003:**
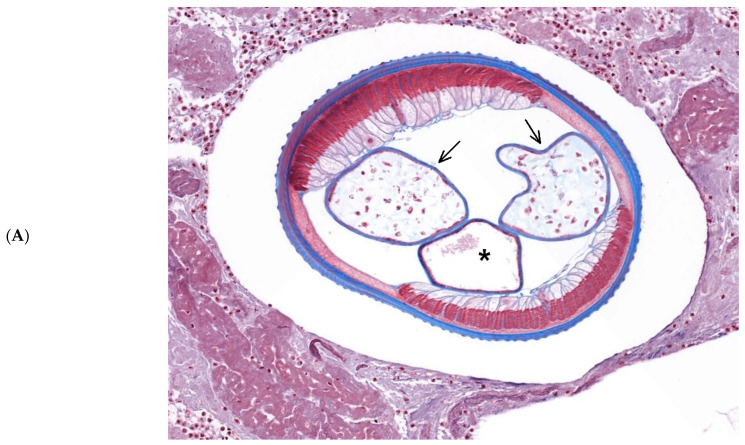
Histological sections of the body of the worm. (**A**): The cross-section shows the digestive tube (asterisk) and the paired uterus (arrows), indicating a female worm (Masson’s Trichrome 100×). (**B**): The outer wall is composed of a thick cuticle with typical longitudinal grooves (arrow). Underneath the cuticle are visible bands of muscle cells (asterisk). Parts of the uterus are also visible (hashtag) (Masson’s Trichrom 400×).

## Data Availability

Not applicable.

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
