# Peer review of "Unusual Masquerading Infraorbital Mass—A Case Report of Human Ocular Dirofilariosis Found in A Ukrainian Patient in Switzerland"

_pathogens, 2023, doi:10.3390/pathogens12080982_

Round 1

Reviewer 1 Report

The clinical case is interesting for publication but I must make a number of important points.

Firstly, the title is misleading. It is a case of a Ukrainian person diagnosed in Switzerland. Calling it the first reported case makes it sound like it is autochthonous. I would title it "... report in Switzerland..." and not first report. Or also "A Unusual clinical presentation of human Dirofilaria repens infection in an Ukrainian patient in Switzerland".

Throughout the text, species are not italicised. In addition, the vector species are not well written. It is not Aedes, it is Aedes spp. and the rest is the same.

In figure 2, I would put only the B and with an arrow I would say where the bulge is located.

Figures 1 and 4 are surprising. The first one looks like a man and then Figure 4 looks like a woman. Likewise. In Fig. 1 the person has wrinkles and the photograph is well focused, and in Fig. 4, the photograph does not look the same as Fig. 1. They have to be photographed with the same camera and under the same conditions. It is therefore recommended to resubmit Figure 4 in the same conditions as Figure 1.

It is not stated how the worm has been stained. It is necessary to know this.

We are talking about a Dirofilaria worm. The species must be stated throughout.

It should be written "dirofilariosis", not "Dirofilariasis".

In the discussion, nothing is said about the situation of dirofilariosis in Ukraine, which is the country of origin of the person studied, as he has only been in Switzerland for 6 months.

The references are appropriate and necessary as they stand.

Author Response

Firstly, the title is misleading. It is a case of a Ukrainian person diagnosed in Switzerland. Calling it the first reported case makes it sound like it is autochthonous. I would title it "... report in Switzerland..." and not first report. Or also "A Unusual clinical presentation of human Dirofilaria repens infection in an Ukrainian patient in Switzerland".

Title modified

Throughout the text, species are not italicised. In addition, the vector species are not well written. It is not Aedes, it is Aedes spp. and the rest is the same.

Text modified

In figure 2, I would put only the B and with an arrow I would say where the bulge is located.

Changes made

Figures 1 and 4 are surprising. The first one looks like a man and then Figure 4 looks like a woman. Likewise. In Fig. 1 the person has wrinkles and the photograph is well focused, and in Fig. 4, the photograph does not look the same as Fig. 1. They have to be photographed with the same camera and under the same conditions. It is therefore recommended to resubmit Figure 4 in the same conditions as Figure 1.

Unfortunately, we do not have any better post-operative photos, which is why I am forced to delete photo 4

It is not stated how the worm has been stained. It is necessary to know this.

It’s Masson’s Trichrome staining and this information has been added to the text

We are talking about a Dirofilaria worm. The species must be stated throughout.

The text in the article has been changed

It should be written "dirofilariosis", not "Dirofilariasis".

The text in the article has been changed

In the discussion, nothing is said about the situation of dirofilariosis in Ukraine, which is the country of origin of the person studied, as he has only been in Switzerland for 6 months.

information added to the discussion with reference

Reviewer 2 Report

Dear authors, very clear clinical case. Well presented. I agree with you that it should be Dirofilaria repens, but only with morphology based on histology section, it is impossible to differentiate from Dirofilaria immitis, and very difficult from Onchocerca vulpis. I would may be add something on the discussion about the probability very high, rather than confirmation 100%.

All latin names in italic.

You are using both dirofilariosis and dirofilariasis. Personally I prefer using the international scientific nomenclature which is "osis". but "iasis" is tolerated and massively used by medical doctors. Nevertheless, try to homogeneize and avoid both in the manuscript.

Author Response

Dear authors, very clear clinical case. Well presented. I agree with you that it should be Dirofilaria repens, but only with morphology based on histology section, it is impossible to differentiate from Dirofilaria immitis, and very difficult from Onchocerca vulpis. I would may be add something on the discussion about the probability very high, rather than confirmation 100%.

the text of the article has been modified

All latin names in italic.

the text of the article has been modified

You are using both dirofilariosis and dirofilariasis. Personally I prefer using the international scientific nomenclature which is "osis". but "iasis" is tolerated and massively used by medical doctors. Nevertheless, try to homogeneize and avoid both in the manuscript.

the text of the article has been modified with dirofilariosis

Reviewer 3 Report

The bibliographic citations should be improved

Author Response

References have been improved

Round 2

Reviewer 1 Report

Thank you for accepting the suggestions

Author Response

Thank you for your valuable comments.